# Antioxidant Properties of Lyophilized Rosemary and Sage Extracts and its Effect to Prevent Lipid Oxidation in Poultry Pátê

**DOI:** 10.3390/molecules25215160

**Published:** 2020-11-06

**Authors:** Mirelli Bianchin, Daiane Pereira, Jacqueline de Florio Almeida, Cristiane de Moura, Rafaelly Simionatto Pinheiro, Leila Fernanda Serafini Heldt, Charles Windson Isidoro Haminiuk, Solange Teresinha Carpes

**Affiliations:** Department of Chemistry, Federal University of Technology—Paraná (UTFPR), P.O. Box 591, 85503-390 Pato Branco, Brazil; bianchin.mirelli@gmail.com (M.B.); daiannepereiraa@hotmail.com (D.P.); jack_florio@hotmail.com (J.d.F.A.); tiane.moura@hotmail.com (C.d.M.); rafaellysimionatto@hotmail.com (R.S.P.); leilafernandaserafini@gmail.com (L.F.S.H.); haminiuk@utfpr.edu.br (C.W.I.H.)

**Keywords:** *Rosmarinus officinalis*, *Salvia officinalis*, natural antioxidant, antibacterial activity, total phenolic compounds, HPLC, thiobarbituric acid reactive substances (TBARS)

## Abstract

This study aimed to evaluate the antioxidant activities (AA) of lyophilized rosemary extract and lyophilized sage extract, and their effects on the oxidative stability of poultry pátê. For this purpose, four poultry pátê formulations with rosemary, sage, sodium erythorbate and a control (without antioxidants) were produced. The rosemary and sage were characterized according to total phenolic compounds (TPC) and AA by several methods. The poultry pátês stored at 4 °C were evaluated by the lipid oxidation. High concentrations of TPC were detected in the rosemary extract and sage extract (46.48 and 41.61 mg GAE/g (Gallic acid equivalent), respectively). The AA of the rosemary and sage extracts determined by free radical-scavenging were 4745.72 and 2462.82 µmol TE/g (Trolox equivalents), respectively. The high concentrations of catechin, rutin, myricetin and p-coumaric acids in these extracts may be responsible for the strong inhibitory action against food pathogens. Besides this, these compounds can be responsible for the best performance in inhibiting lipid oxidation in poultry pátês during storage. This study suggests that rosemary and sage extracts may be used as a natural antioxidant in meat products.

## 1. Introduction

Rosemary (*Rosmarinus officinalis*) and sage (*Salvia officinalis*) are plant of the Lamiaceae family, and both are native to the Mediterranean, however they grow in many parts of the world and have adapted easily in Brazil. Several researchers have reported that rosemary and sage demonstrate excellent antioxidant potential [1,2]. The antioxidant activity of rosemary can be attributed to the presence of phenolic compounds, mainly rosmarinic acid, carnosol, rosmanol, isorosmanol, rosmadiol and carnosic acid [3], while rosmarinic acid and derivatives of caffeic acid are mainly responsible for the effect of the antioxidant activity of sage extracts. Rosemary and sage essential oils have been used in several types of food, such as mayonnaise, meat products, pie dough and salad dressing, as a natural antioxidant. In fact, the rosemary essential oil and lyophilized rosemary extract exhibited an excellent inhibitory effect on the lipid oxidation of pork sausage, with a high acceptance rate [2]. In another study, Pereira et al. [1] reported that rosemary extract had strong anti-oxidative effects in chicken burgers at 21 days of storage at refrigerated temperatures. In addition, according to Estevéz et al. [4], a mix of sage and rosemary essential oils inhibited the oxidative deterioration of liver pâtés to a greater extent than butylatedhydroxytoluene (BHT) did.

The consumption of chicken meat has increased in recent years, and there is a tendency to substitute the use of red meat for white meat, due to the low cost of chicken meat when compared to that of other animal proteins. However, chicken meat is rich in iron, and has several proteins indispensable to humans. Additionally, chicken meat is also a source of energy and other nutrients, such as B vitamins (niacin and riboflavin), minerals and lipids. In 2019, a total of 5.81 billion chickens were slaughtered on the Brazilian market. The states of the southern region together slaughtered 60.6% of the total national slaughter, more than double the regional shares of the Southeast (19.9%), Midwest (13.9%), Northeast (3.8%) and North (1.6%). The state of Paraná in 2019 led the national ranking of chicken slaughter, with more than 94.52 million chickens slaughtered, constituting >32.5%, followed by Santa Catarina and Rio Grande do Sul [5].

Pâté is defined by Brazilian legislation as a pasty product, obtained from meat, meat products and edible offal, and added with food ingredients. The paste must be subjected to an appropriate thermal process and can be cooked, pasteurized or sterilized [6]. In these products, lipid oxidation is one of the main reactions that can occur and can affect the nutritional quality of the food products. These oxidative processes in meat products are influenced by polyunsaturated fatty acids in the presence of prooxidating agents, such as iron, oxygen, salt, and even mechanical processes. However, these effects can be reduced or inhibited by the use of antioxidants in meat products [7]. In fact, the use of synthetic antioxidants to extend the shelf life of meat and meat products is common in food technology. However, the synthetic antioxidants used in the meat industry, such as butylatedhydroxyanisole (BHA), butylatedhydroxytoluene (BHT), tert-butylhydroquinone (TBHQ) and propyl gallate (PG), have been restricted by many countries due to possible risks to consumer health [8]. The toxicological effects of these chemical additives are drawing interest onto studies of natural substances with antioxidant activities in meat [9].

The main mechanism of action of phenolic compounds from plants is the inactivation of lipid free radicals [10]. However, the reduction in the production of reactive oxygen species (ROS) and, consequently, the interruption of the propagation phase of lipid autoxidation, can also occur. Thus, this work aimed to determine the antioxidant properties and total phenolic compounds of lyophilized rosemary extract and lyophilized sage extract, and the effects of their addition in poultry pátês to avoid lipid oxidation during refrigerated storage at 4 °C.

## 2. Results and Discussion

### 2.1. Total Phenolic Compounds, Total Flavonoids, and Preliminary Chromatography Analysis and Antioxidant Activity

The quantification of total phenolics and total flavonoids offers insight into how promising the sample’s bioactive properties are, since they are related to antioxidant activity. In fact, phenolic compounds from fruits, vegetables and medicinal herbs are presumed to be a good source of antioxidants. The total phenolic compounds and total flavonoids present in the extracts of rosemary and sage are shown in Table 1. There was a statistical difference between the extracts for total phenolic compounds and total flavonoids (*p* < 0.05). The rosemary extract had the highest content of phenolic compounds (46.48 mg EAG/g of plant), and this result is in agreement with a previous study by Bianchin et al. [2], who measured 45.67 mg EAG/g for the lyophilized rosemary extract that was extracted with ethanol solution in a shaker at 40 °C for 60 min. However, in this study, the sage extract showed superiority in the content of flavonoids (15.03 mg QE/g of plant) compared with the lyophilized rosemary extract (11.89 mg QE/g). Our results showed that the total phenolic content of lyophilized sage extract was lower than the values reported by Kivrak et al. [11], who found, for total phenolic compounds, 52.3 mg of pyrocatechol/g in the methanolic extract of sage from Turkey. These differences between the TPC and TFC quantities obtained in our study and found in the literature can be attributed to the different methods of extraction, solvents and temperatures employed.

Despite the importance of TPC and TFC analysis, and despite being widely used in plant extracts, individual phenolic compound analysis is very important and necessary in order to know which compounds may be responsible for the antioxidant activity, or even which compounds may be involved in the possible synergistic action of these extracts. Thus, the individual phenolic compounds in both the extracts were determined by HPLC/DAD, and are shown in Table 2.

Among seven phenolic acids and flavonoids identified in the rosemary extracts, the gallic acid was present with the largest mass fraction (3.45 mg/100 g) in phenolic acids, followed by ferulic acid (3.21 mg/100 g) and p-coumaric acid (1.37 mg/100 g), while rutin (34.62 mg/100 g) had the largest mass fraction among flavonoids, followed by catechin (13.21 mg/100 g), quercetin (11.56 mg/100 g), and kaempferol (2.24 mg/100 g) in the rosemary extract. In this study, the same phenolic acids found in rosemary extract were found also in the sage extract. Nonetheless, p-coumaric acid (30.10 mg/100 g) was the most abundant acid phenolic found in the sage extracts. In the sage extract, it was possible to identify five flavonoids—among them the myricetin showed the highest concentration (166.62 mg/100 g), followed by rutin (6.16 mg/100 g), kaempferol (4.93 mg/100 g) and quercetin (0.62 mg/100 g). Although both plants are from the Lamiaceae family and the same region of collection, no myricetin was found in the rosemary extract, while catechin was not found in the sage extract (Table 2).

### 2.2. Antioxidant and Antibacterial Activity

Pátê is a manufactured meat product which, due to its high-fat, is predisposed to deterioration by lipid oxidation, and also microbial contamination due to high water content. However, this degradation can be minimized, and the use of natural antioxidants can increase the shelf life of the meat products. The antioxidant activities of these based-plant extracts that are rich in phenolic compounds may act at different stages in the oxidation process, lowering the free radical concentration, chelating ions and leading to non-radical compounds [12]. The structure of the phenolic compound facilitates the electron donation of the hydroxyl portion to the oxidizing radical species.

Accordingly, the antioxidant activity of the rosemary extract and sage extract was assessed by four distinct in-vitro methods, including the free radical scavenging ability as assessed by the use of ABTS, DPPH radical, ferric reducing antioxidant power (FRAP), and oxidation of β-carotene/linoleic acid. These results are shown in Table 1. These four spectrophotometric methods are complementary and have different mechanisms of action, which can depend on the reaction of an organic radical, a cation radical, or a complex with an antioxidant molecule capable of donating a hydrogen atom [12]. Besides, the absorbance can be measured to test the amount of iron reduced, and can be correlated with the amounts of antioxidants in several matrices. All this enabled us to report complete information on the antioxidant capacity of these samples. Regarding the antioxidant activity assessed by the DPPH method, expressed as EC_50_, it is important to highlight that the EC_50_ value is inversely proportional to the antioxidant capacity of the compounds.

Given all these considerations, the rosemary and sage extracts showed EC_50_ values of 254.72 and 398.86 µg/mL, respectively (Table 1). Based in our findings, the rosemary extract showed antioxidant activities lower than those reported by Bianchin et al. [2], who found 127.33 µg/mL (EC_50_) in the extract of rosemary leaves. In addition, Hamrouni-Sellami et al. [13] found values of EC_50_ = 23.87 µg/mL for the sage methanolic extract from Tunisia. Several studies evaluating the antioxidant activity of Salvia species can be found in the literature, although each study presents a different methodology and expresses its results in many ways. Besides, many of these studies are on sage species originating from different regions where the soil and the climate interfere in the results.

As to the antioxidant activity assessed by the DPPH method, expressed as Trolox (TE), the ethanolic extracts from rosemary and sage showed 4745.72 and 2462.82 µmol TE/g contents, respectively. The ability of the polyphenols and other compounds present in the biological cells to reduce the ferric ion to ferrous ion has been used to evaluate the antioxidant activity in food matrices [14]. The results obtained from the ABTS, FRAP and β-carotene/linoleic methods for rosemary extract were 301.47 µmol TE/g, 180.09 µmol Fe^2+^/g and 85.67%, respectively, and are shown in Table 1. Except for the antioxidant activity determined by the β-carotene/linoleic acid model system, all these values of antioxidant activity for rosemary extract were higher than those reported for sage.

The antibacterial activities of the extracts of rosemary and sage against two strains of Gram-positive and two strains of Gram-negative bacteria are shown in Table 3. Although Gram-negative bacteria are usually more resistant to several bioactive compounds due to the hydrophilic surface of their outer membrane [15], in this study, the rosemary extract showed a strong inhibitory activity against *Staphylococcus aureus* with an MIC of 0.15 mg/mL, and a slightly lower inhibition capacity against *Salmonella enteritidis* (MIC = 0.30 mg/mL). Besides this, the tested *Bacillus cereus* strains seemed to be more sensitive to both extracts, rosemary and sage. In fact, at higher concentrations, the rosemary extract was able to inhibit *Klebsiella pneumoniae* and *Bacillus cereus* with MIC values of 0.63 and 1.25 mg/mL, respectively. Nevertheless, in the tested rosemary extract concentrations, no bacteria showed bactericidal action (MBC > 5 mg/mL) (Table 3). Similarly, Abramovič et al. [16] found in rosemary leaf extracts a stronger antimicrobial activity at 0.02 mg/mL for Gram-positive and 1.03 mg/mL for Gram-negative bacteria. The observed antibacterial activity of rosemary can be associated with its phytochemical composition, such as the flavonoids, and mainly the catechin, rutin and quercetin, according to the data highlighted in the HPLC analysis. These flavonoids are described as potent antibacterial agents against several types of food pathogens [16]. The rutin is described as a strong inhibitor of *Escherichia coli*, Klebsiella sp, and *Pseudomonas auruginosa* [17]. Additionally, the rutin seems to have an action that is synergistically enhanced with other flavonoids against *Bacillus cereus* and *Salmonella enteritidis*. In another study, the addition of rutin decreased the minimum inhibitory concentration value for kaempferol [18]. Additionally, the high concentration of catechin in the rosemary extract may be responsible for the strong inhibitory action against all the pathogenic bacteria tested in this study. In fact, catechin and epigolocatechin gallate from teas were able to inhibit and kill some Gram-positive and Gram-negative bacterial species [19].

Though the ethanolic extract of sage did not inhibit the *Staphylococcus aureus* at the concentrations tested in this study (MIC > 5 mg/mL), it did inhibit the bacteria *Samonella enteritidis* and *Bacillus cereus* at higher concentrations of 0.63 mg/mL and 1.25 mg/mL, respectively. Nevertheless, the sage extract at a concentration of 1.25 mg/mL was able to destroy the cells of *Klebsiella pneumoniae* during 24 h incubation periods, indicating the bactericidal effect of this extract. In this study, the myricetin was the most abundant flavonol in the sage extract (166.82 mg/100g), and Cetin-Karaca and Newman [20] demonstrated the antibacterial effect of the myricetin, rutin, quercetin and catechin against *Escherichia coli* and *Salmonella enteritidis*. Besides the antibacterial activity, these phenolic compounds have revealed other biological activities, such as antioxidant, antidiabetic, anticancer, immunomodulatory, cardiovascular, analgesic and antihypertensive [21].

It is known that plants have enormous chemical complexity in their structure, and the diversity of compounds present in each of these plants may be related to the biological activity of these compounds, whether the action is in isolation or synergistic. In fact, individual chemical characteristics, such as the presence of the phenolic compounds in these extracts, can be responsible for these activities. Besides, phenolic acids can also interact synergistically with flavonoids or other compounds, increasing the antioxidant and antibacterial activities.

### 2.3. Effect of Rosemary and Sage Extracts on Lipid Oxidation from Poultry Pátês

The oxidative degradation of lipids is one of the main reactions that occur in food products, and can be inhibited by the use of natural antioxidants in several kinds of food products. These changes are the main cause of the short shelf-life of these foods, and the oxidative changes of the meat products can be monitoring by the 2-thiobarbituric acid reactive substances test (TBARS). This test is the most common method, and is generally used as an indicator of the degree of lipid oxidation in meat products. It reflects the content of malondialdehyde (MDA) formed during the oxidation of polyunsaturated fatty acids, which are secondary products of oxidation. Thus, the results obtained in the evaluation of lipid oxidation by TBARS in the four treatments are shown in Table 4.

During the storage period of the pátês, the MDA values increased over time, and ranged from 1.35 ± 0.05 at the beginning of the experiment to 2.87 ± 0.06 mg MDA/Kg pâté at the end of the experiment (28 days). Regarding the control treatment, the TBARS average values ranged from 1.66 to 3.96 mg MDA/Kg pâté, and had significantly higher TBARS values (*p* < 0.05) than other treatments. Besides, there was not a significant difference in the malonaldehyde values when comparing treatments with sage extracts and with sodium erythorbate (T2, T3) on the processing day. The treatment of pátês with rosemary extract (T1) showed lower TBARS values during any day of storage than the sage extract- (T2), sodium erythorbate- (T3) and control (T2)-treated pátês. Bilska et al. [22] observed similar results in liver pátês with the addition of rosemary extract.

At the end of the storage time (28 days), the poultry pátês containing rosemary extract (T1) and sage extract (T2) inhibited, respectively, 43.69% and 32.07% of lipid oxidation in comparison to the control pátês (T4). On this point, the treatment with rosemary was better than the treatment with synthetic antioxidant, and the pátês group with sage extract was not significantly different (*p* < 0.05) from the treatment group with synthetic antioxidant (T3).

These results indicate a strong anti-oxidative effect of lyophilized rosemary extract and sage extract in poultry pátês, probably due to the gradual release of the bioactive compounds from these plants. Rutin, catechin and quercetin were present in rosemary extract in high concentrations, and may be responsible for the best inhibition of lipid oxidation observed in pátês containing rosemary extract when compared to pates added with the sage extract. Besides, although p-coumaric acid and myricetin are present in high concentrations in the sage extract, the absence of catechin may have a more significant influence in reducing the inhibition of lipid oxidation in the pátês. Nevertheless, the pate with sage showed no statistical differences from the formulation containing the synthetic antioxidant, sodium erythorbate. A slowing down in the formation of malonaldehyde during lipid oxidation also was observed when the rosemary extract was added in other products, such as Cantonese sausage [23], pork sausage [2], flaxseed oil [24] and white pan bread [25].

## 3. Materials and Methods

### 3.1. Chemicals

Sodium erythorbate (SE), potassium acetate, ethanol, ethylenediaminetetraacetic acid (EDTA), thiobarbituric acid (TBA) and chloroform were obtained from Vetec (Sao Paulo, Brazil). Quercetin, rutin, ferulic acid, coumaric acid, pinocembrin, DPPH (2,2-diphenyl-1-picrylhydrazyl), ABTS (2,2′- azino-bis (3-ethylbenzothiazoline-6-sulfphonic acid), TPTZ (2,4,6-Tris(2-pyridyl)-s-triazine), Folin–Ciocalteu phenol reagents, Trolox, β-carotene/linoleic acid, were obtained from Sigma–Aldrich (Sternheim, Germany).

### 3.2. Material and Preparation of the Extracts

Rosemary (*Rosmarinus officinalis*) and sage (*Salvia officinalis*) samples were purchased at a farmers fair in Pato Branco, Paraná, Brazil. Rosemary and sage leaves were dried in an oven with forced air circulation (400/D model, Nova Ética Ind., São Paulo, Brazil) at 40 °C for 42 h and ground in an analytical mill (less than 0.5 mm).

Samples containing 10 g (dry basis) were subjected to the extraction process with 100 mL of ethanol solution (800 mL/L) in a water-bath at 70 °C for 30 min at a stirring rate of 150 rpm. The extract was filtered through qualitative filter paper, and the supernatants were evaporated in a rotary evaporator (vacuum pressure of 600 mm Hg and 40 °C) until completely dry and lyophilized (L101 model, Liobras Ind., São Carlos, Brazil).

### 3.3. Preliminary Chromatographic Analyses of Phenolic Compounds

The analysis of phenolic compounds in the rosemary and sage extracts through high-performance liquid chromatography was carried out using a Dionex Ultimate 3000 (Dionex, Idstein, Germany) system chromatograph, including a column (Acclaim^®^ 120 C18) at 40 °C with a photodiode array detector (HPLC/PDA). The mobile phases were carried out according to Pereira et al. [1] and the data analyses were processed with Chromeleon software. Ethe extracts were filtered through a 0.22 μm filter nylon syringe (Millipore, São Paulo, Brazil) and the injection volume was 10 μL with a flow rate of 1.0 mL/min. The total run time was 55 min and the spectral data were collected from 280 to 370 nm. The identification was performed by comparison of retention times and absorption in ultraviolet. The quantification was performed by external standardization [26]. The following authentic standards of phenolic compounds were examined: ferulic acid, gallic acid, chlorogenic acid, caffeic acid, p-coumaric acid, trans-cinnamic acid, syringic acid, kaempferol, myricetin, rutin, and quercetin. The contents of the bioactive compounds in the sample were expressed as mg/100g. Determination by HPLC was also performed in triplicate.

### 3.4. Total Phenolic Compounds (TPC) and Total Flavonoids Content (TFC)

The total phenolic content (TPC) was performed using the Folin–Ciocalteu method described by Singleton et al. [27] using gallic acid as the standard. After two hours in darkness, the absorbance of the extract was measured at 764 nm in a spectrophotometer UV-VIS (Bel Spectro 2000, Biovera, Rio de Janeiro, Brazil). The results are expressed as mg GAE/g of sample (GAE: gallic acid equivalent). The total flavonoid content (TFC) was quantified by the colorimetric method with aluminum chloride [2]. After 40 min at room temperature, the absorbance was measured at 415 nm in a spectrophotometer. A quercetin standard curve was obtained, and the results were expressed as Quercetin equivalent (mg QE/g). All the assays were carried out in triplicate.

### 3.5. DPPH (2,2 Diphenyl-1-Picryl-Hydrazyl) Radical Scavenging Assay

DPPH (2,2-diphenyl-1-picryl-hydrazyl) radical scavenging was performed according to the methodology described by Brand-Williams [28]. The mixture was incubated at room temperature in the dark for 45 min and the absorbance was read using a spectrophotometer UV-VIS (Bel Spectro 2000, Biovera, Rio de Janeiro, Brazil) at 517 nm. The result was expressed in µmol TE/g (Trolox equivalent). Additionally, the EC_50_ (concentration required to obtain a 50% antioxidant effect) values were calculated by means of a linear regression between the concentration in µg/mL (axis of the abscissas) and the mean percentage of antioxidant activity (ordinal axis).

### 3.6. Ferric Reducing Antioxidant Power (FRAP) Assay

The FRAP was determined as described in Pulido et al. [29]. The reaction mixture was incubated for 30 min in water bath at 37 °C, and the absorbance was measured at 595 nm. Aqueous solutions of ferrous sulfate were used for calibration, and the results were expressed as μmol of Fe^2+^/g.

### 3.7. ABTS (2,2-Azino-Bis-(3-Ethylbenzothiazoline-6-Sulphonic Acid) Assay

The ABTS (2,2′-Azino-bis (3-ethylbenzthiazoline-6-sulphonic acid)) method was performed as described by Re et al. [30] in the rosemary and sage extract. The stock solutions included 7.4 mM ABTS^°+^ and 2.6 mM potassium persulfate. The solution was diluted by mixing 1 mL ABTS^°+^ solution with 60 mL ethanol to an absorbance of 0.70 ± 0.02 units at 734 nm on a spectrophotometer (UV 2000, Femto Ind., São Paulo, Brazil). Antioxidant activity was expressed in μmol/g of TE (Trolox equivalent).

### 3.8. Coupled Oxidation of β-Carotene and Linoleic Acid Assay

The measure of antioxidant activity was determined by the coupled oxidation of β-carotene and linoleic acid. Emulsion oxidation was spectrometrically monitored (Bel Spectro 2000, Biovera, Rio de Janeiro, Brazil) by measuring its absorbance at 470 nm, at time zero (t = 0) and subsequently after every 20 min, until the characteristic color of β-carotene disappeared in the control reaction (t = 100 min). The antioxidant activity was determined as the percent inhibition relative to the control sample [1].

### 3.9. Minimum Inhibitory Concentration (MIC) and Minimum Bactericidal Concentration (MBC)

The in vitro potential antibacterial activities of the rosemary and sage extracts were qualitatively estimated based on the broth micro-dilution test in 96-well sterile plates, employing four bacterial strains [31]. The *Staphylococcus aureus* ATCC 25.923 and the *Bacillus cereus* ATCC 11.778 as Gram-positive bacteria, and *Salmonella enteritidis* ATCC 13.076 and *Klebsiella pneumoniae* ATCC 13.883 as the Gram-negative bacteria, were used for the antibacterial examinations of aforementioned herbs. The strains were reactivated from stock cultures in liquid BHI medium (Brain Heart Infusion) for 24 h at 37 °C, and later grown on BHI agar plates. A bacterial suspension with NaCl 0.89% sterile of 1–2 × 10^8^ UFC/mL on the Mc Farland scale was obtained using the absorbance of 0.135 at 660 nm. A concentration of 1–2 × 10^5^ UFC/mL was obtained by adding 50 µL of the bacterial suspension into 50 mL of BHI broth. In the microplate, the MIC was evaluated with 190 µL of inoculated BHI and 10 µL of each extract in concentrations of 0.15 to 5.000 mg/mL. The chloramphenicol 0.12% was used as the positive control, the ethanol as negative control and the resazurin dye was employed to reveal the positive wells. The plates were maintained for 24 h at 37 °C. The experiment was performed three times and the results are presented in mg/mL of the extract. For the MBC test, an aliquot of each positive well in MIC was inoculated into a Petry dish containing BHI agar and incubated for 24 h at 37 °C. The MBC was considered the lowest concentration without visible microbial growth.

### 3.10. Poultry Pâté Elaboration and Oxidative Stability of Pâtés

The rosemary and sage extracts were applied in poultry pâté to evaluate the effect on the inhibition of the lipid oxidation of the product, based on the preliminary studies with chicken pâtés [7]. The pâté formulation was prepared with drumstick and leg (330 g/Kg), breast (80 g/Kg) and chicken fat (246 g/Kg). The cuts of the thigh, drumstick, breast and fat were ground twice through a 5 mm plate at −2 °C before the application of the other ingredients. Of this mixture, 70% was cooked slowly at 60 ± 3 °C, in order to soften the connective tissue. The raw and cooked paste were mixed and then the remaining ingredients were added: ice water (274 g/Kg), salt (13 g/Kg), a mix of dehydrated garlic, onion, chives, parsley and pepper (4 g/Kg), isolated soy protein (6.0 g/Kg), cassava starch (40 g/Kg), sodium polyphosphate (5 g/Kg), and carmine (0.73 g/Kg). This basic formulation was divided into treatments. T1 was added 2 g/Kg of lyophilized rosemary extract, T2 was added 2 g/Kg of lyophilized sage extract, T3 was added sodium erythorbate (SE) (0.2 g/kg), and T4 had no additional ingredients. Separately, the pâté samples were stored in sterile glasses, autoclaved and submitted to a baking process in a water bath at 80 °C for 30 min. After baking, the pâtés were cooled with ice and stored at 4 °C in a refrigerator for 28 days to evaluate the oxidative stability of the pâtés. The oxidative stability of poultry pátês was estimated by using 2-thiobarbituric acid reactive substances (TBARS) according to Carpes et al. [7] Tetramethoxypropane (TMP) was used as the standard reference and substances reacting with thiobarbituric acid were measured spectrophotometrically at 532 nm at 0, 7, 14, 21 and 28 days of storage. The results were expressed as mg of MDA/kg of sample (MDA: malondialdehyde). The lipid oxidation was assessed in triplicate.

### 3.11. Statistical Analysis

Analysis of variance (ANOVA) was performed to analyze the data, and the means were compared by Tukey’s test for TBARS values, using the Statistica 8.0 software (Stat Soft Inc., Tulsa, OK, USA). The results were considered statistically significant when *p* < 0.05. All tests were performed in triplicate.

## 4. Conclusions

The present study represents a relevant contribution to the bioactivity knowledge of rosemary and sage grown in Brazil, which are important sources of phenolic compounds with antioxidant activities. Therefore, these results suggest that these plants could be of great industrial importance, and can support the development of natural additives with potential applications in food technology. Besides this, considering the interest in finding natural antioxidants, it is possible to suggest that lyophilized rosemary extract and lyophilized sage extract have biological activity mainly because of the presence of reducing compounds, free radical scavengers and hydrogen donors in the plant material. In fact, the rosemary and sage extract were more effective in lipid oxidation inhibition in poultry pátês than synthetic antioxidants, and these pátês with rosemary and sage could be an alternative for the consumer demand of healthy foods. The antioxidant and antimicrobial activities, and mainly the inhibition of lipid oxidation in foods rich in fats, are some of the attributes expected in the search for a natural and safe food additive.

## Figures and Tables

**Table 1 molecules-25-05160-t001:** Total phenolic content, total flavonoid content and antioxidant activity of rosemary and sage extracts.

Parameters	Rosemary Extract	Sage Extract
Total Phenolic Compound (mg GAE/g)	46.48 ± 0.08	41.61 ± 0.91
Total Flavonoid Content (mg QE/g)	11.89 ± 0.58	15.03 ± 0.44
EC_50_ (µg/mL)	254.72 ± 0.66	398.86 ± 0.10
AA % *	92.25 ± 0.46	79.81 ± 0.56
DPPH (µmol TE/g)	4745.72 ± 0.47	2462.82 ± 0.03
ABTS (µmol TE/g)	301.47 ± 0.76	182.89 ± 0.526
FRAP (µmol de Fe^2+^/g)	180.09 ± 0.01	133.99 ± 0.68
β-carotene/linoleic (%) **	85.67 ± 0.66	89.29 ± 0.10

QE: Quercetin equivalent; EC_50_: Equivalent concentration required to obtain a 50% antioxidant effect; TE: Trolox equivalent; * lyophilized rosemary and sage extract at 0.33 mg/mL; ** lyophilized rosemary and sage extract at 6.66 mg/mL. Values are presented as mean ± deviation (*n* = 3).

**Table 2 molecules-25-05160-t002:** Content of phenolic compounds by HPLC of rosemary and sage extracts.

Phenolic Compounds	Rosemary Extract (mg/100g)	Sage Extract (mg/100g)	RT (min)	λ (nm)
Gallic acid	3.45 ± 0.05 ^d^	0.46 ± 0.01 ^f^	6.65	280
Catechin	13.21 ± 1.05 ^b^	-	10.37	280
p-Coumaric acid	1.37 ± 0.02 ^g^	30.10 ± 0.15 ^b^	13.94	320
Ferulic acid	3.21 ± 0.01 ^e^	0.35 ± 0.03 ^g^	18.57	320
Rutin	34.62 ± 1.15 ^a^	6.16 ± 0.10 ^c^	19.88	370
Myricetin	-	166.62 ± 1.21^a^	27.38	320
Quercetin	11.56 ± 0.50 ^c^	0.62 ± 0.02 ^e^	31.50	320
Kaempferol	2.24 ± 0.01 ^f^	4.93 ± 0.31 ^d^	34.32	370

RT: Retention time; λ: Wave length; Values mean ± standard deviation. Different letters (a–g) in the same column indicate significant differences (*p* < 0.05) by Tukey’s test.

**Table 3 molecules-25-05160-t003:** Antibacterial activity of rosemary and sage extracts.

Bacterial	Rosemary Extract		Sage Extract	
	MIC (mg/mL)	MBC (mg/mL)	MIC (mg/mL)	MBC (mg/mL)
*Staphylococcus aureus*	0.15	>5	>5	>5
*Bacillus cereus*	1.25	>5	1.25	>5
*Salmonella enteritidis*	0.30	>5	0.63	>5
*Klebsiella pneumoniae*	0.63	>5	1.25	>1.25

MIC: Minimum inhibitory concentration; MBC: Minimum bactericidal concentration.

**Table 4 molecules-25-05160-t004:** Average values of TBARS in poultry pâté with rosemary and sage extract during storage at 4 °C.

Treatments	TBARS (mg of MDA/Kg of Chicken Pâté)				
	Storage Times Days				
	0	7	14	21	28
T1	0.98 ± 0.02 ^Dd^	1.23 ± 0.04 ^Cc^	1.87 ± 0.01 ^Bb^	1.88 ± 0.02 ^Bc^	2.23 ± 0.02 ^Ac^
T2	1.18 ± 0.10 ^Ec^	1.71 ± 0.04 ^Db^	1.86 ± 0.01 ^Cb^	2.37 ± 0.03 ^Bb^	2.69 ± 0.02 ^Ab^
T3	1.58 ± 0.04 ^Eb^	1.73 ± 0.05 ^Db^	2.00 ± 0.01 ^Ca^	2.28 ± 0.03 ^Bb^	2.60 ± 0.04 ^Ab^
T4	1.66 ± 0.03 ^Ea^	1.85 ± 0.02 ^Da^	1.91 ± 0.03 ^Ca^	2.87 ± 0.02 ^Ba^	3.96 ± 0.30 ^Aa^
Average	1.35 ± 0.05 ^E^	1.63 ± 0.04 ^D^	1.91 ± 0.02 ^C^	2.35 ± 0.04 ^B^	2.87 ± 0.06 ^A^

Values are presented as mean ± deviation (*n* = 3). MDA: malondialdehyde. All measurements were carried out in triplicate. T1: Treatment added 0.2 g/Kg of lyophilized rosemary extract; T2: Treatment added 2 g/Kg of lyophilized sage extract; T3: Treatment with sodium erythorbate (0.2 g/Kg); T4: Treatment control (no added antioxidant); Different lower-case letters (a–d) in the same column indicate significant differences (*p* < 0.05) by Tukey’s test. Different capital letters (A–E) in the same row indicate significant difference (*p* < 0.05) by Tukey’s test.

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
