# Peer review of "Antioxidant Properties of Lyophilized Rosemary and Sage Extracts and its Effect to Prevent Lipid Oxidation in Poultry Pátê"

_molecules, 2020, doi:10.3390/molecules25215160_

Round 1
Reviewer 1 Report
Natural antioxidants are widely used in the food, cosmetic and pharmaceutical industries. The antioxidants from these plants have been extensively researched. The composition and antioxidant properties of the extracts were analyzed once again.
The extracts do not possess bactericidal activity in the dose used in the processing of pate (2 mg / g), but inhibiting the growth of bacteria is possible. It would be interesting in this case to investigate the bacterial contamination of the pate at the end of the experiment or to control the infection of the pate with the studied bacteria. In addition, it was interesting to know the change in the organoleptic properties of the pate during such processing.
The manuscript will be of interest to food technologists and scientists working with natural antioxidants.
Some remarks
18 and sage extracts by free radical-scavenging were 4745.72 and 2462.82 μmol Trolox/g, respectively.
Have to be
18 and sage extracts by free radical-scavenging were 4745.72 and 2462.82 μmol TE/g (Trolox equivalents), respectively.
25 phenolic compounds; HPLC; TBARS.
Have to be
25 phenolic compounds; HPLC; thiobarbituric acid reactive substances (TBARS).
92 Table 1.
Have to be
DPPH (μmol TEAC /g)
ABTS (μmol TEAC/g)
The same standard is used in two tests, abbreviation and dimension must be the same.
281 at 517 nm. The result was expressed in μmol Trolox/g. Additionally, the EC50 (concentration required
Have to be
281 at 517 nm. The result was expressed in μmol TEAC/g. Additionally, the EC50 (concentration required
Trolox itself is not analyzed in the test, but its equivalent activity only.
311 1-2x105 UFC.mL-1 was obtained adding 50 μL of the bacterial suspension into 50 mL of Brain Heart
312 Infusion (BHI) broth. In the microplate, the MIC was evaluated with 190 μL of inoculated BHI and
Have to be
311 1-2x105 UFC.mL-1 was obtained adding 50 μL of the bacterial suspension into 50 mL of
312 BHI broth. In the microplate, the MIC was evaluated with 190 μL of inoculated BHI and
Author Response
Response 1:
Thank you very much for your time and comments that improved the quality of our manuscript. We have modified the manuscript accordingly. All the suggested changes have been highlighted in red in the manuscript file. In addition, detailed corrections are listed below point by point:
.. The extracts do not possess bactericidal activity in the dose used in the processing of pate (2 mg / g), but inhibiting the growth of bacteria is possible. It would be interesting in this case to investigate the bacterial contamination of the pate at the end of the experiment or to control the infection of the pate with the studied bacteria. In addition, it was interesting to know the change in the organoleptic properties of the pate during such processing.
The manuscript will be of interest to food technologists and scientists working with natural antioxidants.
Response: We agree with the reviewer and appreciate the suggestions for completing our work. The concentration of natural extract and also that of synthetic antioxidant was chosen according to the limits recommended by Brazilian legislation. In the next moment we will do sensory analysis, microbiological contamination and other physical chemical attributes to establish a quality and safe product for consumers.
Some remarks
Line18: … and sage extracts by free radical-scavenging were 4745.72 and 2462.82 μmol Trolox/g, respectively.
Have to be
Line18: and sage extracts by free radical-scavenging were 4745.72 and 2462.82 μmol TE/g (Trolox equivalents), respectively.
Response: We apologize for our mistakes. We promptly accepted all the suggestions and made the alterations as recommended. Please see the new version of the manuscript.
Line 25: phenolic compounds; HPLC; TBARS.
Have to be
Line 25: phenolic compounds; HPLC; thiobarbituric acid reactive substances (TBARS).
Response: We apologize for our mistakes. New keywords was added and was highlighted in red in the manuscript.
Line 92: Table 1.
Have to be
DPPH (μmol TEAC /g)
ABTS (μmol TEAC/g)
The same standard is used in two tests, abbreviation and dimension must be the same.
Line 281: … at 517 nm. The result was expressed in μmol Trolox/g. Additionally, the EC50 (concentration required
Have to be
Line 281: … at 517 nm. The result was expressed in μmol TEAC/g. Additionally, the EC50 (concentration required
Trolox itself is not analyzed in the test, but its equivalent activity only.
Response: All the above questions refer to the same subject, so let's answer them together.We agree with the reviewer and this mistake was corrected and the antioxidant activity was expressed in μmol/g of TE (Trolox equivalent) for both methods (ABTS and DPPH). This change was made throughout the text.
Line 311: 1-2x105 UFC.mL-1 was obtained adding 50 μL of the bacterial suspension into 50 mL of Brain Heart
Line 312: Infusion (BHI) broth. In the microplate, the MIC was evaluated with 190 μL of inoculated BHI and
Have to be
Line 311: 1-2x105 UFC.mL-1 was obtained adding 50 μL of the bacterial suspension into 50 mL of
Line 312: BHI broth. In the microplate, the MIC was evaluated with 190 μL of inoculated BHI and ..
Response: We agree with the reviewer and thank the opportunity to explain better. The nw text was added in the text. Please see the new version of the manuscript.
Thank you again for your time and support,
We look forward to hearing from you soon

Reviewer 2 Report
In the manuscript “Antioxidant Properties of Lyophilized Rosemary and Sage Extracts and its Effect to Prevent LIPID Oxidation in Poultry Pátê”, the authors highlighted as lyophilized rosemary and sage extracts can improve the features of a food always more consumed. Today the interest to natural products characterized by biological activities is high, in this case due to important antioxidant and antimicrobial capacities of the molecules that are present in the rosemary and sage. The introduction is good, experimental part is complete with a good statistical analysis. The conclusions are persuasive. For all these reasons, the article is suitable for the publication in this form.
Author Response
Revisor 2
In the manuscript “Antioxidant Properties of Lyophilized Rosemary and Sage Extracts and its Effect to Prevent LIPID Oxidation in Poultry Pátê”, the authors highlighted as lyophilized rosemary and sage extracts can improve the features of a food always more consumed. Today the interest to natural products characterized by biological activities is high, in this case due to important antioxidant and antimicrobial capacities of the molecules that are present in the rosemary and sage. The introduction is good, experimental part is complete with a good statistical analysis. The conclusions are persuasive. For all these reasons, the article is suitable for the publication in this form.
Response: Thank you for your time and support.
Reviewer 3 Report
x
I have reviewed the manuscript titled: Antioxidant properties of lyophilized rosemary and sage extracts and its effect to prevent lipid oxidation in poultry pate. The LIPID in title at first page should revise to Lipid.
This article aims to evaluate the use of lyophilized rosemary extract and sage extract as natural antioxidant on the lipid oxidation comparing to sodium erythorbate and antibacterial effect of chicken pate and to analyze the phenolic compounds of the extracts. The information of this work is useful and relevant and there are seven main phenolic compounds in each extract of the manuscript that could be adapted by poultry processing industry especially for chicken pate shelf-life in the future. I think the manuscript is acceptable after major revision. The article is not innovative, however, it contains original and interesting information for poultry processing of chicken pate. Abstract is well written upon and the total phenolic compounds for rosemary and sage extracts are mentioned and evaluated. Introduction is well addressed including natural antioxidants except antibacterial activity of rosemary and sage extracts is not cited and miss-spelling reactive species at oxygen (ROS) as “oxigen”. The information of pate in Brazil was introduced and why the rosemary and sage extracts were used in this study.
Materials and methods were well described.
This article would be improved if the authors identify eight phenolic compounds by LC/MS.
I am not a native English speaker. The manuscript seems have no major mistakes are detected and the manuscript can be easily understood. The results are well discussed.
References
There are at least 20 references not follow the required format for Molecules. Therefore, it is not acceptable before major revision as attached file.
I enjoyed reading this manuscript; the needs of special groups of poultry processing of chicken pate and other species high in lipid. This manuscript presents some interesting data.
Date of this review
27 October 2020 9:14

Author Response
Response: Thank you very much for your time and comments that improved the quality of our manuscript. We have modified the manuscript accordingly. All the suggested changes have been highlighted in red in the manuscript file.
I have reviewed the manuscript titled: Antioxidant properties of lyophilized rosemary and sage extracts and its effect to prevent lipid oxidation in poultry pate. The LIPID in title at first page should revise to Lipid.
Response: The word LIPID in the title was corrected and the and the capital letter was removed.
.. This article aims to evaluate the use of lyophilized rosemary extract and sage extract as natural antioxidant on the lipid oxidation comparing to sodium erythorbate and antibacterial effect of chicken pate and to analyze the phenolic compounds of the extracts. The information of this work is useful and relevant and there are seven main phenolic compounds in each extract of the manuscript that could be adapted by poultry processing industry especially for chicken pate shelf-life in the future. I think the manuscript is acceptable after major revision. The article is not innovative, however, it contains original and interesting information for poultry processing of chicken pate. Abstract is well written upon and the total phenolic compounds for rosemary and sage extracts are mentioned and evaluated. Introduction is well addressed including natural antioxidants except antibacterial activity of rosemary and sage extracts is not cited and miss-spelling reactive species at oxygen (ROS) as “oxigen”. The information of pate in Brazil was introduced and why the rosemary and sage extracts were used in this study. Materials and methods were well described. This article would be improved if the authors identify eight phenolic compounds by LC/MS. I am not a native English speaker. The manuscript seems have no major mistakes are detected and the manuscript can be easily understood. The results are well discussed.
Rsponse:
Line 66: oxigen
Response: We agree with the reviewer and thank the opportunity to corrected it.
New word was added. oxygen. In addition, We agreed that additional chromatographic techniques could be used, however at this first moment we opted for a faster and more available technique in our laboratory. This work will continue and we can include the suggested analyzes.
References
There are at least 20 references not follow the required format for Molecules. Therefore, it is not acceptable before major revision as attached file.
I enjoyed reading this manuscript; the needs of special groups of poultry processing of chicken pate and other species high in lipid. This manuscript presents some interesting data.
Response: References were checked and corrected according to the Journal 's rules. Please see the new version of the manuscript.
Thank you very much for your time and comments that improved the quality of our manuscript.
Round 2
Reviewer 3 Report
There are still three small mistakes as the attached file should be revised before accepting the manuscript.

This manuscript is a resubmission of an earlier submission. The following is a list of the peer review reports and author responses from that submission.